# Group Structure as a Foundation for Entropies

**DOI:** 10.3390/e26030266

**Published:** 2024-03-18

**Authors:** Henrik Jeldtoft Jensen, Piergiulio Tempesta

**Affiliations:** 1Centre for Complexity Science and Department of Mathematics, Imperial College London, South Kensington Campus, London SW7 2AZ, UK; 2Department of Computer Science, School of Computing, Tokyo Institute of Technology, 4259, Nagatsuta-cho, Yokohama 226-8502, Japan; 3Departamento de Fisica Teórica, Universidad Complutense de Madrid, 28040 Madrid, Spain; p.tempesta@fis.ucm.es; 4Instituto de Ciencias Matemáticas (ICMAT), 28049 Madrid, Spain

**Keywords:** entropy, composability, extensivity, information theory, power laws, group theory

## Abstract

Entropy can signify different things. For instance, heat transfer in thermodynamics or a measure of information in data analysis. Many entropies have been introduced, and it can be difficult to ascertain their respective importance and merits. Here, we consider entropy in an abstract sense, as a functional on a probability space, and we review how being able to handle the trivial case of non-interacting systems, together with the subtle requirement of extensivity, allows for a systematic classification of the functional form.

## 1. Introduction

The term “entropy” is used extensively in the modern scientific literature. Originating in the 19th-century theory of thermal dynamics [1], the concept is now, to a near bewildering extent, used widely in one form or another across many sciences. For example, entropy is at the foundation of information theory [2] and is of crucial use in computer science [3,4]. Also, neurosciences make use of entropy both as a tool to characterize and interpret data from brain scans [5] and, more fundamentally, in theories of the dynamics of the brain and mind [6]. Generally speaking, entropy is a fundamental notion in complexity science [7]. Here, we present a brief review of some recent mathematical developments in the theory of entropy.

In mathematical terms, an entropy is a functional S[p] defined on a space of probability distributions $p=(p_{1},p_{2},\cdots ,p_{W})$ associated with a *W*-dimensional event space. Thus, we use the word “entropy” with the same meaning it assumes, for instance, in the case of Rényi’s entropy, without direct reference to thermodynamics. From this perspective, the relevance of entropies is clear. They can be considered as analytic tools that can help in the analysis of the inter-dependencies within the system behind a given event space. Similarly, their use in information-theoretic analysis of time series is likewise natural. The connection between entropies as mathematical functionals and the thermodynamic entropy of Clausius defined in terms of heat transfer is much less immediate. Here, we will concentrate on the mathematical aspects of entropies as functionals and only make a few comments on the possible connection to thermodynamics.

The first question to tackle is which functional form of S[p] yields useful entropies. i.e., how to limit the infinite number of choices for S[p]. It is well known that the Boltzmann–Gibbs–Shannon form
(1)$$S_{\mathrm{BGS}}\left[p\right]=\sum_{i=1}^{W}p_{i}\log\frac{1}{p_{i}}$$
(we assume $k_{B}=1$) is the unique possibility if one assumes that the entropy must satisfy the four Shannon–Kinchin (SK) axioms [8]:(SK1)(Continuity). The function $S(p_{1},\ldots ,p_{W})$ is continuous with respect to all its arguments.(SK2)(Maximum principle). The function $S(p_{1},\cdots ,p_{W})$ takes its maximum value over the uniform distribution $p_{i}=1/W,i=1,\ldots ,W$.(SK3)(Expansibility). Adding an event of zero probability to a probability distribution does not change its entropy: $S\left(p_{1},\ldots ,p_{W},0\right)=S\left(p_{1},\cdots ,p_{W}\right)$.(SK4)(Additivity). Given two subsystems A, B of a statistical system, S(A∪B)=S(A)+S(B|A).

Therefore, to derive entropies of a functional form different from the one in Equation (Equation 1), it is necessary to go beyond the four SK axioms. Various strategies in this respect have been adopted in the literature.

Let us start by recalling Constantino Tsallis’s elegant observation [9] that the formula
(2)$$S_{q}\left[p\right]=k\frac{1-\sum_{i=1}^{W}p_{i}^{q}}{q-1},\;\;\;\;p=\left(p_{1},p_{2},\cdots ,p_{W}\right)\in {\left[0,1\right]}^{W},\;\;\;k\in R_{+}$$
provides a possible generalization of Boltzmann’s entropy. This is the case in the sense that $S_{q}$ is a functional on the space of probability distributions $p:\left\{1,2,\cdots ,W\right\}\mapsto {\left[0,1\right]}^{W}$, and in the limit q→1, the entropy $S_{q}\left[p\right]$ becomes equal to the Boltzmann–Gibbs–Shannon entropy in Equation (Equation 1). Tsallis’s 1988 article [9] has inspired a tremendous effort to generalize Boltzmann’s entropy in different scenarios, including what we will review in this paper. Tsallis pointed out that the entropy $S_{q}$ fulfills a specific procedure for combining independent systems which can be seen as a generalization of the additivity property (SK4) of the Boltzmann–Gibbs–Shannon entropy. In particular, Tsallis suggested that the free parameter *q* should be determined by requiring that $S_{q}$ for a given system is extensive (for a recent reference to Tsallis’s argument, see [10]), i.e., that in the uniform case where the probabilities are $p_{i}=1/W$ for all i=1,2,⋯,W, the entropy $S_{q}\propto N$ for n→∞, where *N* denotes the number of components in the system under analysis. For clarity, we note that when considering physical systems, the entropy may become volume-dependent; for example, because the number of states available, *W*, depends on the volume. Volume dependence can also enter through the probabilities $p_{i}^{*}$ determined by the maximum entropy principle.

Although the Tsallis entropy does not fulfill axiom SK4 in its original form and hence is *non-additive*, it does satisfy a composition relation different from addition.

Another set of non-Boltzmann–Gibbs–Shannon entropies was derived by Hanel and Thurner [11] by simply discarding axiom SK4 and then determining the functional form from the asymptotic behavior of the number of events, or states, *W*, as a function of the number of components in the system. However, in this approach, there is no systematic rule for handling the computation of the entropy of a system consisting of independent parts.

It is well known that in physics, the investigation of required symmetries has often been helpful. Think of Einstein’s use of the symmetry between different reference frames to derive special and general relativity theory. Consider also the eightfold way and the derivation of QCD. Additionally, consider the application of symmetry and group theory to atomic spectra. Therefore, it seems natural to, rather than discarding the fourth SK axiom, replace it in a way informed by the symmetry properties (and related group-theoretic restrictions) that an entropy must necessarily satisfy. Consequently, the question is, which symmetry cannot be ignored when dealing with entropies?

The fourth SK axiom addresses how the entropy of a system AB, consisting of two independent parts *A* and *B*, can be expressed as a sum of the entropy of *A* and the entropy of *B*. Tsallis entropy also allows the entropy of AB to be expressed in terms of the entropy of the two parts, not as a sum but as a generalized combination of the entropies of the parts.

The notion of group entropy, introduced in [12,13], goes one step further and exploits the idea that the process of combining two independent systems can be seen as a group operation. The group entropies satisfy the three first SK axioms, as well as a fourth one, which consists of a generalized composition procedure making use of formal group theory [14,15].

This approach leads to axiomatically defined entropies, whose composition rule is defined in terms of the generator G(t) of a suitable formal group law, namely
(3)$$S\left[p\right]=\frac{G\left(\ln\sum_{i=1}^{W}p_{i}^{\alpha }\right)}{1-\alpha }\;\;\mathrm{with}\;\alpha >0\;\mathrm{and}\;\alpha \neq 1.$$

Although this restricts the allowed functional forms available for an entropy, it does not uniquely determine the entropy as the four SK axioms do. Below, we will discuss how the analysis of combining independent systems using formal group theory, together with requiring extensivity, allows for a systematic classification of entropies in “universality classes”. These classes are defined by taking into account how fast the number of available states *W* grows with the number of components *N*. We accomplish this by starting with a more general functional non-trace form than the one given in Equation (Equation 2). For details, see [12].

By generalizing the Tsallis functional form and requiring composability together with extensivity, we are able to regard Tsallis entropy as the composable entropy associated with systems where interdependence between their components forces *W* to grow slowly, namely as a power of *N*. Below, we will discuss why composability on the entire probability space is an indispensable property of an entropy. We will also address the need for extensivity in a general sense. We will point out that extensivity can be very relevant even beyond the thermodynamic need for a well-defined limit as the number of components approaches infinity. For example, extensivity is essential for using an entropy as a measure of the complexity of a time series or for handling power-law probability distributions.

## 2. Why Composability

The need for composability does not arise because real systems can always be regarded as a simple combination of subsystems. The requirement is, in a sense, a logical necessity [16]. When we consider two independent systems with state spaces *A* and *B*, we should obtain the same result if we compute the entropy of the Cartesian combined system A×B as if we first compute the entropy of *A* and *B* separately and then afterward decide to consider them as one combined system. By Cartesian combination, we mean that the system A×B is given by the set of states {(a,b)|a∈A,b∈B}, with the probabilities for the individual states given by p(a,b)=p(a)p(b). This Cartesian combination immediately suggests familiar properties from group theory. The composition is as follows:Commutative: Since we have state spaces in mind, we consider A×B=B×A. The ordering is immaterial.Associative: A×(B×C)=(A×B)×C.“Neutral” element: A×B∼A if B={b}. In other words, A×B is essentially the same set as *A* if *B* consists of one element only. In terms of state spaces, all sets with one state only are considered to be identical, that is, indistinguishable. In this sense, a unique “neutral” element exists in our composition process. Accordingly, we want the entropy of a probability distribution on a set containing a single element to be zero: indeed, it would correspond to a certainty configuration. Moreover, we want the entropy of A×B to be equal to the entropy of *A* if the entropy of *B* is zero.

The group structure of the Cartesian combination of the event spaces for systems must also be satisfied by the entropy functional operating on the corresponding probability spaces. This can be ensured by employing formal group theory [16]. Define the entropy using the expression in Equation (Equation 3), where the function $G\left(t\right)=t+\sum_{k=2}^{\infty }\beta_{k}t^{k}$ is the “group generator”. Here, the formal power series G(t) is said to be the group exponential in the formal group literature [15]. The combination of independent systems is now expressed as
(4)S(A×B)=ϕ(S(A),S(B)),
where the function ϕ(x,y) is given by $\phi \left(x,y\right)=G(G^{-1}\left(x\right)+G^{-1}\left(y\right))$. Given a formal group law, namely when ϕ(x,y) is a formal power series in two variables, it is possible to prove that there exists a one-variable formal power series ψ(x) such that ϕ(x,ψ(x))=0. This series represents the formal inverse and completes the group structure.

## 3. Why Extensivity

Let us first recall the definitions of extensivity and additivity. We say that an entropy is extensive if the entropy per component is finite in the limit of infinitely many components, i.e.,
(5)$$\lim_{N\to \infty }S\left(N\right)/N=\mathrm{constant}<\infty$$
An entropy is additive if, for two statistically independent systems *A* and *B*, the entropy of the two systems considered as a combined system is equal to the sum of the entropies, i.e.,
(6)S(A+B)=S(A)+S(B).

When considering the thermodynamics of macroscopic systems with the number of constituents, *N*, of the order of Avogadro’s number, the usual procedure is to compute quantities such as the thermodynamic free energy *F* for an arbitrary value of *N*. The limit of large, essentially infinite systems is then handled by considering intensive quantities, e.g., the free energy per constituent. Hence, for an entropy to be thermodynamically useful, it needs to be extensive, given the fundamental thermodynamic relation F=E−TS. Since the temperature *T* is an intensive quantity, the entropy must be extensive. Thus, we need the limit $\lim_{N\to \infty }S\left(N\right)/N$ to be well defined.

Outside thermodynamics, entropy finds a significant application within information theory as a tool to characterize the complexity of a deterministic or random process generating a time series. More precisely, we can associate with a time series an ordinal representation formed by all ordinal patterns of length L∈N assigned [17]. Assuming that all different patterns are allowed for a process, we have W(L)=L!, and each pattern *i* will occur with a probability of $p_{i}=1/L!$. The Boltzmann–Gibbs–Shannon entropy in Equation (Equation 1) is given by $S_{\mathrm{Shan}}\left[p\right]=\ln L!≃L\ln L-L$. So, we obtain a diverging entropy rate S[p]/L as the length of the time series increases. As we will see, this is a common situation since random processes exhibit super-exponential growth in the number of permitted patterns. Again, extensivity enters into play. Thus, we would need an entropy that grows proportionally to the number of allowed patterns in the considered time series.

The widespread occurrence of power-law probability distributions in nature, either exact or approximate, has long been the focus of self-organized criticality (for an overview, see [18,19]). It is now clear that power-law distributions with fat tails are common, and for this reason, it seems natural to consider the extent to which the workhorse of information theory, the Shannon entropy, can be used as a meaningful entropic measure for such distributions.

Consider a probability distribution of the following form:(7)$$P_{S}\left(s\right)=\frac{A}{s^{a}}\;\;\mathrm{for}\;\;s=1,2,\cdots ,s{\left(N\right)}_{\max}.$$
Here, *A* is a normalization factor and *a* is a positive exponent. The variable *s* denotes the “size” of some process, e.g., an avalanche or a structure such as a spatial cluster. When $s{\left(N\right)}_{\max}$ grows with *N*, the usual Boltzmann–Gibbs–Shannon entropy will, in general, not allow a well-defined limit $S\left[P_{S}\right]\left(N\right)/N$ as N→∞.

## 4. The Structure of the Group Entropies

Here, we explain why the expression in Equation (Equation 3) is a good starting point for deriving generalized entropies. First, we address why we choose the argument of the generating function G(t) to be $\ln\sum_{i}p_{i}^{\alpha }$. We also comment on the so-called trace form of the group entropies given by
(8)$$S\left[p\right]=\sum_{i=1}^{W}p_{i}G\left(\ln\frac{1}{p_{i}}\right).$$

Finally, we briefly recapitulate how the functional form of G(t) is determined by reference to formal group theory and the requirement that S[p] is extensive on the uniform (also denoted as the microcanonical) ensemble, given by
(9)$$p_{i}=\frac{1}{W(N)},\;\;\mathrm{for}\;\;i=1,2\cdots ,W\left(N\right).$$

The structure of Equation (Equation 3) is used as the starting point because G(t) being a group generator ensures composability for all distributions $p_{i}$, not only the uniform distributions. And, taking the argument to be $\ln\sum_{i}p_{i}^{\alpha }$ enables this functional form to generate a range of well-known entropies, including Boltzmann–Gibbs–Shannon, Rényi, and Tsallis [12]. More specifically, if one chooses
(10)$$G\left(t\right)=\frac{e^{at}-e^{bt}}{\left(a-b)(\alpha -\beta \right)}$$
one recovers the Boltzmann–Gibbs–Shannon entropy in the limit α→1; Rényi’s entropy in the double limit a→0,b→0; and Tsallis’s entropy in the double limit a→1,b→0.

### 4.1. Extensivity and the Group Law G(T)

Let us now briefly describe how the requirement of extensivity determines the group law G(t) in Equation (Equation 3). Details can be found in [20,21]. For a given dependence of the number of available states W(N), we want to ensure that the entropy given in Equation (Equation 3) is extensive, i.e., that on the uniform ensemble $p_{i}=1/W\left(N\right)$ for i=1,⋯,W(N) we have $\lim_{N\to \infty }S\left[p\right]/N=\mathrm{constant}$.

We can express this as
(11)$$S\left(p_{i}=\frac{1}{W}\right)=\lambda N.$$
Asymptotically, we have
(12)$$S\left(\frac{1}{W}\right)=\frac{G(\ln\left(W^{1-\alpha }\right)}{1-\alpha }\approx \lambda N.$$
Then, we invert the relation between *S* and *G*, which, by Equation (Equation 12), amounts to inverting the relation between *G* and *N*. For G(t) to generate a group law, we must require G(0)=0 [12,16], so we adjust the expression for G(t) accordingly and conclude that
(13)$$G\left(t\right)=\lambda \left(1-\alpha \right)\{W^{-1}\left[\exp\left(\frac{t}{1-\alpha }\right)\right]-W^{-1}\left(1\right)\}.$$
Hence, given the asymptotic behavior of W(N), we derive different corresponding entropies. In the expressions below, $\lambda \in R_{+}$, α>0, and α≠1 are free parameters.


*Non-trace-form case:*



(I)Algebraic, $W\left(N\right)=N^{a}$
(14)$$S\left[p\right]=\lambda \left\{\exp\left[\frac{\ln(\sum_{i=1}^{W(N)}p_{i}^{\alpha })}{a(1-\alpha )}\right]-1\right\}.$$(II)Exponential, $W\left(N\right)=k^{N}$
(15)$$S\left[p\right]=\frac{\lambda }{\ln k}\frac{\ln(\sum_{i=1}^{W(N)}p_{i}^{\alpha })}{1-\alpha }.$$This is, of course, the Rényi entropy.(III)Super-exponential, $W\left(N\right)=N^{\gamma N}$
(16)$$S\left[p\right]=\lambda \left\{\exp\left[L\left(\frac{\ln\sum_{i=1}^{W(N)}p_{i}^{\alpha }}{\gamma (1-\alpha )}\right)\right]-1\right\}.$$This entropy was recently studied in relation to a simple model in which the components can form emergent paired states in addition to the combination of single-particle states [22].


So far, we have only considered the so-called non-trace form of the group entropies given in Equation (Equation 3). A set of entropies can be constructed in the same manner, starting with the trace-form ansatz in Equation (Equation 8).

### 4.2. Trace-Form Group Entropies

It is interesting to observe that the ansatz in Equation (Equation 8) directly leads to either the Boltzmann, the Tsallis, or an entirely new entropy, depending on the asymptotic behavior of W(N). By applying the procedure described in Section 4.1, we obtain the following three classes corresponding to the ones considered for the non-trace case.


*Trace-form case:*



(I)Algebraic, $W\left(N\right)=N^{a}$
(17)$$\begin{array}{cc} S[p] & =\lambda \sum_{i=1}^{W(N)}p_{i}\left[{\left(\frac{1}{p_{i}}\right)}^{\frac{1}{a}}-1\right] \end{array}$$
(18)$$\begin{array}{cc} & =\frac{1}{q-1}\left(1-\sum_{i=1}^{W(N)}p_{i}^{q}\right). \end{array}$$To emphasize the relation with the Tsallis q-entropy, we have introduced q=1−1/a and λ=1/(1−q). Note that the parameter *q* is determined by the exponent *a*, so it is controlled entirely by W(N).(II)Exponential, $W\left(N\right)=k^{N}$, k>0
(19)$$S\left[p\right]=\frac{\lambda }{\ln k}\sum_{i=1}^{W(N)}p_{i}\ln\frac{1}{p_{i}}.$$This is the Boltzmann–Gibbs–Shannon entropy.(III)Super-exponential, $W\left(N\right)=N^{\gamma N}$, γ>0
(20)$$S\left[p\right]=\lambda \sum_{i=1}^{W(N)}p_{i}\left\{\exp\left[L(-\frac{\ln p_{i}}{\gamma })\right]-1\right\}.$$


### 4.3. Examples of Systems and Corresponding Group Entropies

To illustrate the classification of group entropies based on the asymptotic behavior of W(N), we consider three Ising-type models:(a)The Ising model on a random network [11].(b)The usual Ising model, for example, with nearest-neighbor interaction on a hyper-cubical lattice.(c)The so-called pairing model in which Ising spins can form paired states [22].
Let *E* denote the total energy of the system. We are interested in the asymptotic behavior of the number of possible states for the three models as a function of *N* for fixed energy per component ϵ=E/N. First, consider (a). As explained in [11], $W\left(N\right)∼N^{a}$ when the fraction of interaction links, the connectance, in the considered network, is kept constant as the number of nodes *N* is increased. The exponent *a* is given by the ratio between the energy density and the connectance. The entropy corresponding to this functional form of W(N) is, for all values of the exponent *a*, given by the Tsallis entropy [21].

The entropy corresponding to the standard Ising model (case (b)) with $W\left(N\right)=2^{N}$ is the Boltzmann–Gibbs–Shannon entropy. The pairing version of the Ising model (case (c)) admits a super-exponential growth in the number of states $W\left(N\right)∼N^{\gamma N}$, leading us to a new functional form of the entropy [22]
(21)$$S_{\gamma ,\alpha }\left[p\right]=\exp\left[L\left(\frac{\ln\sum_{i=1}^{W}p_{i}^{\alpha }}{\gamma (1-\alpha )}\right)\right]-1.$$

## 5. Group Entropies and the Ubiquity of the Q-Exponential Distribution

It is well known that the q-exponential form relating to the Tsallis q-entropy provides a very good fit to an impressively broad range of data sets (see, e.g., [10]). This may, at first, appear puzzling given that we saw in Section 4.1 that the Tsallis entropy corresponds to one of the three classes considered here, namely systems with strong interdependence between the components that $W\left(N\right)∼N^{a}$. The reason that the q-exponential appears to be much more pervasive than one would expect, given that the q-entropy is restricted to the case $W\left(N\right)∼N^{a}$, may be due to the following.

Consider the maximum entropy principle. For all the classes of entropies considered in Section 4, the probability distribution that maximizes the entropy is a q-exponential. The probability distribution for the specific case of $W\left(N\right)∼k^{N}$ is the usual exponential Boltzmann distribution. But since the Boltzmann distribution is the limiting case of the q-exponential for q→1, we can say that, independently of the asymptotic behavior of W(N), the maximum entropy principle always leads to q-exponential distributions [21].

How can this be? The reason is the functional form of the argument
$$x\equiv \ln\sum_{i=1}^{N}p_{i}^{\alpha }$$
of the ansatz in Equation (Equation 3). When one applies Lagrange multipliers and extremizes the entropy in Equation (Equation 3), the q-exponential functional form will arise from the derivative $\partial x/\partial p_{i}$. The remaining factors in the expression for the derivative of S[p] will depend on the functional form of the group law G(t) but will formally just be a constant if evaluated on the maximizing distribution $p^{*}$ and do not depend explicitly on $p_{i}$.

## 6. An Entropic Measure of Complexity

Fully interacting complex systems possess a number of microstates W(N) that may be different from the Cartesian exponential case $W\left(N\right)=\Pi_{i=1}^{N}k_{i}$, where $k_{i}$ is the number of states available to component number *i* in isolation. When interactions freeze out states, W(N) can grow slower than exponentially with increasing *N*. In contrast, when interactions allow for the creation of new states from the combination of components, W(N) can grow faster than exponentially. As an example, think of hydrogen atoms that form hydrogen molecules $H+H\to H_{2}$. The states of $H_{2}$ are not just the Cartesian product of free single hydrogen atomic states.

The possible difference between the cardinality of the state space of the fully interacting system and the state space formed as a Cartesian product of the states available to the individual components can be used to construct a new measure of the degree of emergent interdependence among the components of a complex system. We can think of this as a quantitative measure of the degree of complexity in a given system. We imagine the entire system AB to be divided into two parts, *A* and *B*, and compare the entropy of the system A×B, obtained by combining the micro-sates of the two parts as a Cartesian product, with the system AB, obtained by allowing full interaction between the components of *A* and those of *B*. We denote by AB this fully interacting system. The complexity measure is given by [20]
(22)Δ(AB)=S(A×B)−S(AB)=ϕ(S(A),S(B))−S(AB).

From the dependence of Δ(AB) on the number of components in the separate systems *A* and *B*, one can, in principle, determine the kind of emergence generated by the interactions in a specific complex system. In [20], we conjectured that the number of available states for the brain grows faster than exponentially in the number of brain regions involved. It might, at first, appear impossible to check this conjecture. However, experiments like those conducted on rat brain tissue, such as the famous avalanche experiment by Beggs and Plentz [23], seem to open up the possibility for a study of Δ(AB) as a function of tissue size. We imagine it would be possible to study a piece of tissue of size *N* and one of size 2N, allowing, at least in principle, to determine how Δ(AB) behaves for such a neuronal system. A different, although related, notion of complexity, the defect entropy, was proposed in [24].

## 7. Group Entropy Theory and Data Analysis

The theory of group entropies has recently proved to be relevant in data analysis. One important reason for this relevance is extensivity. When the number of patterns that may occur in a given time sequence depends, in a non-exponential way, on the length *L* of the sequence, the Shannon-based entropy of the sequence S(L) will not permit a well-defined entropy rate S(L)/L because the Shannon entropy will not be extensive in *L*. This may, for example, pose a problem for the widely used Lempel–Ziv [25] complexity measure. This is similar to the discussion above concerning how the Boltzmann-Gibbs-Shannon entropy fails to be extensive on state spaces that grow non-exponential in the number of constituents. We will see below that time series very often contain a number of patterns that grow super-exponentially in the length of the sequence.

To discuss this fundamental application of group entropies, we start with a brief review of the ordinal analysis of time-series data. We follow the discussion and notations in [17,26,27]. Consider the time series
$${\left(x_{t}\right)}_{t\geq 0}=x_{0},x_{1},\ldots ,x_{t},\cdots$$
where *t* represents a discrete time and $x_{t}\in R$. Let L≥2. We introduce the sequence of length *L* (or *L*-sequence)
$$x_{t}^{L}:=x_{t},x_{t+1},\cdots ,x_{t+L-1}$$
Let $\rho_{0},\rho_{1},\ldots ,\rho_{L-1}$ be the permutation of 0,1,…,L−1 such that
(23)$$x_{t+\rho_{0}}<x_{t+\rho_{1}}<\ldots <x_{t+\rho_{L-1}}\;.$$
We denote the rank vector of the sequence $x_{t}^{L}$ as follows:(24)$$r_{t}:=\left(\rho_{0},\rho_{1},\ldots ,\rho_{L-1}\right),$$
The rank vectors

$r_{t}$ are called *ordinal patterns* of length *L* (or *L*-ordinal patterns). The sequence $x_{t}^{L}$ is said to be “of type” $r_{t}$. In this way, given the original time series ${\left(x_{t}\right)}_{t\geq 0}$, we have constructed an *ordinal representation* associated with it: the family of all ordinal patterns ${\left(r_{t}\right)}_{t\geq 0}$ of length *L*.

We denote by $S_{L}$ the group of the L! permutations of 0,1,…,L−1, which represents the set of symbols (also called “alphabet”) of the ordinal representation.

In the following, we consider discrete-time stationary processes $X={\left(X_{t}\right)}_{t\geq 0}$, both deterministic and random, taking values in a closed interval I⊂R. We define a “deterministic process” as a “one-dimensional dynamical system” (I,B,μ,f), where *I* is the state space (a bounded interval of R), B is the “Borel σ-algebra” of *I*, μ is a “measure” such that μ(I)=1, and f:I→I is a μ-invariant map. In this case, the image ${\left(X_{t}\right)}_{t\geq 0}$ of *X* is the orbit of $X_{0}$, i.e., ${\left(X_{t}\right)}_{t\geq 0}={\left(f^{t}\left(X_{0}\right)\right)}_{t\geq 0}$, where $f^{0}\left(X_{0}\right)=X_{0}\in I$ and $f^{t}\left(X_{0}\right)=f\left(f^{t-1}\left(X_{0}\right)\right)$. First, we associate with an ordinal representation the probability p(r) of finding an ordinal pattern of a given rank $r\in S_{L}$. To this aim, we assume the *stationary condition*: for k≤L−1, the probability of $X_{t}<X_{t+k}$ cannot depend on *t*. This condition ensures that estimates of p(r) converge as data increase. Non-stationary processes with stationary increments, such as the fractional Brownian motion and the fractional Gaussian noise, satisfy the condition above.

### 7.1. Metric and Topological Permutation Entropy

Let X be a deterministic or random process taking real values. Let p(r) be the probability of a sequence $X_{t}^{L}$ generated by X being of type r, and let p=p(r) be the corresponding probability distribution. We define the following:(i)If p(r)>0, then r is a permitted pattern for *X*.(ii)If p(r)=0, then r is a forbidden pattern.
The *permutation metric entropy of order L* of *p* is defined as
(25)$$H^{*}\left(X_{0}^{L}\right)=-\sum_{r\in S_{L}}p\left(r\right)\ln p\left(r\right).$$
The topological entropy of order *L* of the finite process $X_{t}^{L}$, $H_{0}^{*}\left(X_{t}^{L}\right)$ is the upper limit of the values of the permutation metric entropy of order *L*. Formally, we obtain it by assuming that all allowed patterns of length *L* are equiprobable:(26)$$H_{0}^{*}\left(X_{t}^{L}\right):=\ln A_{L}\left(X\right),$$
where $A_{L}\left(X\right)$ is the number of allowed patterns of length *L* for X. It is evident that the following inequalities hold:$$H^{*}\left(X_{0}^{L}\right)\leq \ln A_{L}\left(X\right)\leq \ln L!$$
We observe that $A_{L}\left(X\right)=L!$ if all *L*-ordinal patterns are allowed.

### 7.2. Bandt–Pompe Permutation Entropy

In their seminal paper [28], Bandt and Pompe introduced the following notions:Permutation metric entropy of X:
$$h_{M}\left(X\right):=\underset{L\to \infty }{lim sup}\frac{1}{L}H^{*}\left(X_{0}^{L}\right)=-\underset{L\to \infty }{lim sup}\frac{1}{L}\sum_{r\in SL}p\left(r\right)\ln p\left(r\right),$$
where $X_{0}^{L}=X_{0},\cdots ,X_{L-1}$.Topological permutation entropy of X:
$$h_{T}\left(X\right):=\underset{L\to \infty }{lim sup}\frac{1}{L}H_{0}^{*}\left(X_{0}^{L}\right)=\underset{L\to \infty }{lim sup}\frac{1}{L}\ln N_{L}\left(X\right).$$

An important question is, what is the relationship between the permutation metric entropy and the standard *Kolmogorov–Sinai (KS) entropy* of a map?

Let *f* be a strictly piecewise monotone map on a closed interval I⊂R. The vast majority of, if not all, one-dimensional maps used in concrete applications belong to the class of piecewise monotone maps. In [29], it was proved that
$$h_{M}\left(f\right)=h_{\mathrm{KS}}\left(f\right).$$

The same relation holds for the topological versions of the two entropies. This is a fundamental result since it allows us to compute the KS entropy using the ordinal analysis approach. The above theorem and its generalizations imply that the number of permitted patterns of length L for a deterministic process grows exponentially as *L* increases.
$$\left|L-\mathrm{permitted}\;\mathrm{patterns}\;\mathrm{for}\;\mathrm{deterministic}\;X=f\right|∼e^{\tau (f)L},$$
where τ(f) is the topological KS entropy. In turn, this implies that the number of prohibited patterns grows super-exponentially.

At the other extreme, we have random processes without prohibited patterns. An elementary example is *white noise*. According to Stirling’s formula,
$$\left|L-\mathrm{possible}\;\mathrm{patterns}\;\right|=L!∼e^{L\ln L}.$$
It is also worth noting that noisy deterministic time series may not have prohibited patterns.

For example, in the case of dynamics on a non-trivial attractor where the orbits are dense, observational white noise will “destroy” all prohibited patterns, regardless of how little the noise is.

In general, *random processes exhibit super-exponential growth in permitted patterns*.

Random processes can also have prohibited patterns. In this case, the growth will be “intermediate,” meaning it is still super-exponential but subfactorial.

### 7.3. A Fundamental Problem

For random processes without prohibited patterns, the permutation entropy diverges in the limit as L→∞:$$h_{T}\left(X\right)=\lim_{L\to \infty }\frac{1}{L}\ln\left|L-\mathrm{possible}\;\mathrm{permitted}\;\mathrm{patterns}\right|=\lim_{L\to \infty }\frac{1}{L}\ln L!=\lim_{L\to \infty }\ln L=\infty$$
Also, in general, $h^{M}\left(X\right)=\infty$. Therefore, it is natural to consider the problem of extending the notion of permutation entropy to make it an intrinsically finite quantity. We assume that for a random process X,
$$\left|L-\mathrm{possible}\;\mathrm{permitted}\;\mathrm{patterns}\;\mathrm{for}\;X\right|∼e^{g(L)}.$$
where g(L) is a certain function that depends on the type of process considered.

Can we find a suitable, generalized permutation entropy that converges as L→∞?

### 7.4. Group Entropies and Ordinal Analysis

We can obtain a new solution to this problem through the theory of group entropies.

**Philosophy**: Instead of using the Shannon-type permutation entropy introduced by Bandt and Pompe as a universal entropy valid for all random processes, we will adapt our entropic measure to the specific problem we wish to address:We will classify our processes into *complexity classes*, defined by complexity functions g(t). These classes, in ordinal analysis, represent a notion entirely analogous to the *universality class* described earlier (inspired by statistical mechanics).Each complexity class will correspond to a *group permutation entropy*, i.e., a specific information measure designed for the class under consideration.This measure will be convergent as L→∞.

#### Functions and Complexity Classes

A process X is said to belong to the complexity class *g* if
$$\ln\underset{A_{L}\left(X\right)}{\underbrace{\left|\mathrm{allowed}\;L-\mathrm{patterns}\;\mathrm{for}\;X\right|}}∼g\left(L\right)\;\mathrm{for}\;L\to \infty .$$
The bi-continuous function g(t) is called the *complexity function* of X. The process X belongs to the *exponential class* if
g(L)=cL(c>0)
X belongs to the *factorial class* if
g(L)=LlnL

**Example** **1.***A deterministic process* X *belongs to the exponential class. A random process*  X *like white noise (*X *i.i.d.) belongs to the factorial class.**A process* X *belongs to the subfactorial class if one of the following conditions holds:***(i)** g(L)=o(LlnL)**(ii)** g(L)=cLlnL;with;0<c<1

**Example** **2.***Processes with*$$g\left(L\right)=L\ln^{\left(k\right)}L;;\left(\ln^{\left(k\right)}L\equiv \underset{k\;\mathrm{times}}{\underbrace{\ln\circ \ln\circ \cdots \circ \ln}(L)}\right)\;\mathrm{with}\;k\geq 2$$*belong to subfactorial class (i). Processes with* g(L)=cLlnL*,* 0<c<1 *can also be constructed explicitly.*

### 7.5. Group Permutation Entropy

Main Result: The conventional permutation entropy of Bandt–Pompe can be consistently generalized. According to our philosophy, *the complexity class g “dictates” its associated permutation entropy*, which becomes finite in the limit of large *L*.

**Definition** **1.**
*The group entropy of order L for a process X of class g is*

$$Z_{g,\alpha }^{*}\left(pL\right)=g^{-1}\left(R\alpha \left(p_{L}\right)\right)-g^{-1}\left(0\right)$$

*where $p_{L}$ is the probability distribution of the L-ordinal patterns of $X0^{L}=X_{0},\cdots ,X_{L-1}$ and $R_{\alpha }\left(p_{L}\right)$ is the Rényi entropy. The corresponding topological group entropy of order L is*

$$Z_{g,0}^{*}\left(pL\right)=g^{-1}\left(\ln AL\left(X\right)\right)-g^{-1}\left(0\right)$$

*The group metric permutation entropy is*

$$z_{g,\alpha }^{*}\left(X\right)=\lim_{L\to \infty }\frac{1}{L}g^{-1}\left(R_{\alpha }\left(p_{L}\right)\right)$$

*The topological group permutation entropy is*

$$z_{g,0}^{*}\left(X\right)=\lim_{L\to \infty }\frac{1}{L}g^{-1}\left(\ln A_{L}\left(X\right)\right)$$

*The functions defined in this manner are group entropies. Furthermore, they satisfy the inequalities*

$$0\leq z_{g,\alpha }^{*}\left(X\right)\leq z_{g,0}^{*}\left(X\right)=1\;\forall \alpha >0.$$

*The following are various examples of group permutation entropies:*

(27)
$$\begin{array}{ccc} & \left(a\right) & \;\mathrm{For}\;g_{\exp}\left(t\right)=ct:\;Z_{g_{\exp},\alpha }^{*}\left(pL\right)=\frac{1}{c}R\alpha \left(p_{L}\right)\\ & \left(b\right) & \;\mathrm{For}\;g_{\mathrm{fac}}\left(t\right)=t\ln t:\;Z_{g_{\mathrm{fac}},\alpha }^{*}\left(pL\right)=e^{L[R\alpha \left(p_{L}\right)]}-1\\ & \left(c\right) & \;\mathrm{For}\;g_{\mathrm{sub}}\left(t\right)=ct\ln t\;\left(0<c<1\right):\;Z_{g_{\mathrm{sub}},\alpha }^{*}\left(pL\right)=e^{L[R\alpha \left(p_{L}\right)/c]}-1 \end{array}$$



## 8. Thermodynamics

The application of non-Boltzmann–Gibbs–Shannon entropies to thermodynamics is subtle. We recall that in standard thermodynamics, it is possible to interpret the Lagrange multiplier corresponding to the constraint on the average energy as the inverse of the physical temperature. However, it is not clear if a similar procedure can be adopted for any generalized entropy.

One can certainly derive the probability weights $p_{i}^{*}$ corresponding to the extrema of
(28)$$J=S-\lambda_{1}\left(\sum p_{i}-N\right)-\lambda_{2}\left(\sum_{i}E_{i}p_{i}-E\right)$$
and we can compute the entropy for these weights $S[p_{i}^{*}]$. However, given an arbitrary generalized entropy S, we do not know if, for some physical systems, there exists a relationship between $S[p_{i}^{*}]$ and Clausius’s thermodynamic entropy defined in terms of heat flow. Hence, to us, the relationship between generalized entropies and thermodynamics in the sense of a theory of heat and energy of physical systems, apart from several interesting analogies, remains an open field of research. Detailed discussions concerning the construction of generalized thermostatistics for the case of the Tsallis entropy $S_{q}$ are available, e.g., in the monographs in [30,31].

## 9. Discussion

The group entropy formalism described has the pleasant property that all group entropies arise systematically and transparently from a set of underlying axioms combined with the requirement of extensivity. This approach is in contrast to those adopted to define many of the existing entropies, which, sometimes, are intuitively proposed or justified by axioms that ignore the need for composability. Many of the most commonly used entropies are included and classified within the group theoretic framework.

The use of information measures adapted to the universality classes of systems, which are extensive by construction, looks promising in several application contexts, such as the study of neural interconnections in the human brain, classical and quantum information geometry, and data analysis in a broad sense. We plan to further investigate complex systems with super-exponentially growing state spaces as a paradigmatic class of examples where these new ideas can be fruitfully tested.

## 10. Conclusions

We have reviewed a group-theoretic approach to the classification and characterization of entropies, regarded as functionals on spaces of probability distributions. The theoretical framework proposed is axiomatic and generalizes the set of Shannon–Khinchin axioms by replacing the fourth additivity axiom with a more general composition axiom. Perhaps the most relevant achievement so far is the systematic classification of the multitude of existing entropies in terms of the rate at which the corresponding dimension of the state space grows with the number of components in the system. A related result is a constructive procedure for entropies, which exhibit extensivity on state spaces of any assigned growth rate. In turn, this property triggers the application of group entropies to information geometry and data analysis.

## Data Availability

Data are contained within the article.

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
