# Peer review of "Group Structure as a Foundation for Entropies"

_entropy, 2024, doi:10.3390/e26030266_

Round 1

Reviewer 1 Report

Comments and Suggestions for Authors

This manuscript presents a review of some aspects of generalized entropies.

I obseve that the manuscript has flows in the presentations and the organization. This is also a collection of the authors' previous works,  although in section 7.5 a small application is presented. Articles in the Special Issue must present new results.

The authors put different contents together from different sources. Consequently, I observe that the review is less focused and the aim of the paper becomes unclear.
Indeed, Sec. 7 is a different topic (permutation entropy) from the content in the previous sections. Readers do not clearly see the connection between the content presented in each section.

Thus, I do not recommend this manuscript for publication.

Some concerns:
- The authors regard extensivity as a very relevant property required for systems with infinite components. However, this does not appear in a later presentation.

- The authors should provide a precise definition of extensivity and its differences from additivity.

- The complexity measure in Eq. (20) is given as the difference in entropy. However, it requires firm arguments because there has been a long discussion of the definition in the literature.

- Sec. 7.5 uses Renyi entropy, which raises the question of why the authors presented Tsallis entropy in the previous sections.

- Discussion presents no point derived from the obtained results.

Some minor points:
- On page 2, line 55, parenthesis ( is not closed.

- The parameter alpha should be denoted in Eq. (3).

- In Eq. (3), the entropy should be written with the index alpha.

- On page 4, line 146-147, for W(L)=L!, is the length L always an integer? What is the factorial of temperature T!?

- In Eq. (10), parenthesis in the numerator is missing.

- Eq. (14) and the subsequent equations contain the unspecified symbol L, and the authors should also denote the parameter gamma.

- On page 7, from line 251 to the end of the paragraph, the text is unclearer.

- In the first line of page 11, there is an undefined abbreviation SF.

Reviewer 2 Report

Comments and Suggestions for Authors

In this manuscript, the authors present and discuss a definition of entropy based on the generator of formal group laws. They argue that this definition forms a good starting point for deriving various forms of entropies, based on the choice of the group generating function G.

They propose a generic form for G, which includes the requirements of composability and extensivity of the so-defined entropy. In particular, the authors show that previously known generalized entropies (such as the Rényi entropy, or the Tsallis entropy) can be recovered by considering different types of scaling for the degeneracy, W(N). They illustrate the use of this entropy in the context of complex system analysis (as a measure of complexity in general, and for time series in particular).

The paper is well written, and provides an interesting, general presentation of more “exotic” definitions of entropy than the average reader might be expected to find. In this sense, the submitted manuscript acts as a short review of the approach proposed by the present authors, with a few original results. This means that the paper has the advantage of being compact and understandable. Conversely, the fact that most discussions are limited in size and scope also leaves the reader wanting more information.

In particular, I would have liked to find a more in-depth discussion on the connection with thermodynamics. The authors do refer to monographs on generalized statistical thermodynamics, but it would have been useful to find in the present manuscript examples of when/where generalized entropies might be useful in this context. In the same vein, I would also have appreciated a more thorough discussion on the kind of information that the authors think could be extracted from time series, based on the proposed generalizations of entropy.

Despite these minor drawbacks, I believe that the submitted manuscript forms an interesting entry point into the group-based approach to alternate definitions of entropy and I found the paper suitable for publication in Entropy.

Comments on the Quality of English Language

The English is fine, but there are a few typos throughout the text.

Reviewer 3 Report

Comments and Suggestions for Authors

VIn this article, the authors present a brief review of extensions of the Boltzmann-Gibbs-Shannon entropy based on formal group theory and produce systematically several of what they call group entropies. They tabulate various entropy functionals and discuss a few examples, mainly for data analysis.

Strong points of the article.

• The paper is well organized with a description of the fundamental concepts, definitions, and axioms in the first pages, such as to be comprehensive to readers not working in this field.

• Some examples are presented and references to applications from the real world are given.

Weak points of the article.

• Applications of group entropies to thermodynamics are superficially discussed, and only two references are given to cover this highly important subject.

• The entropy functional may depend not only on the number of particles and total energy but also on volume.

To my opinion, taking into account the above remarks will considerably strengthen this review article.

Spelling remarks

Lines 120-132 : Using capital letters for \phi and \psi what exactly does it mean?

Line 147 : I assume that T should be L.

Line 270 : micro-statestr/review/review/46074051/l7Y1L4ap

Round 2

Reviewer 1 Report

Comments and Suggestions for Authors

I observe only minor text revision and the typos corrections in the text. The major concerns remain unanswered. In this manuscript, sufficient results and discussion that warrant publication in Entropy are lacking.

- Arguments for the necessity of introducing a measure of complexity are almost lacking.
- Then, why do the authors not use Tsallis entropy instead of Renyi in Sec. 7.5?
- No discussion is provided in the Discussionsection. It only mentions the possibility of several applications.
No quantitative arguments to validate the formalism and limitations are given based on the main results.

Some minor points:
- In Eq. (12), the parentheses in the numerator are not closed.
- The parameter gamma remains undenoted.
- Under the limit symbol in Eq. (5), N-> does not make sense.

Comments on the Quality of English Language

There are some errors.

Reviewer 2 Report

Comments and Suggestions for Authors

The authors have addressed my comments in a satisfactory way. 

Reviewer 3 Report

Comments and Suggestions for Authors

Line 57 : capital letter n

Eq. 5 : N\rightarrow \infty